# Impact of hospitalization on nutritional status in persons aged 65 years and over (NUTRIFRAG Study): Protocol for a prospective observational study

Cristina Carretero-Randez[1,2,3], María Isabel Orts-Cortés[4,5,6]*, Margarita Rodríguez-Pérez[7], Víctor Manuel Gonzalez-Chordá[5,8], Eva María Trescastro-López[9], Joan Blanco-Blanco[6,10,11], Jordi Martínez-Soldevila[10,11], Aránzazu Ruiz-Heras-Hera[12,13], Pedro Raúl Castellano-Santana[14,15], Amando Marquez-Sixto[16], Manuela Domingo-Pozo[17], Antonia Inmaculada Zomeño-Ros[18,19], Jesica Montero-Marco[20,21], Marta Charlo-Bernardos[21,22], Joaquín Moncho[23], Ángel Luís Abad-González[24], María Trinidad Castillo-García[25], Ana Belén Sánchez-García[18,26,27], Ana María De Pascual y Medina[28,29,30], Rosa Ana Clement-Santamaría[17], Ascensión Franco-Bernal[2,31], Rafaela Camacho-Bejarano[5,7,32]

1 Nursing Department, Community Health and History of Science Team, University of Alicante, Alicante, Spain, 2 Nursing Research Group: Infection, Inflammation and Chronicity (IIS La Fe), Hospital Universitario y Politécnico La Fe, Valencia, Spain, 3 Pneumology and Intermediate Respiratory Care Unit, Hospital Arnau de Vilanova, Valencia, Spain, 4 Nursing Department, University of Alicante (BALMIS), Alicante Institute for Health and Biomedical Research (ISABIAL, Group 23), Alicante, Spain, 5 Institute of Health Carlos III, Nursing and Healthcare Research Unit (Investén-isciii), Madrid, Spain, 6 (CIBERFES) Institute of Health Carlos III, CIBER of Frailty and Healthy Ageing, Madrid, Spain, 7 Nursing Faculty, Department of Nursing, University of Huelva, Campus El Carmen, Huelva, Spain, 8 Nursing Department, Nursing Research Group (241), Universitat Jaume I, Castellón, Spain, 9 Alicante Institute for Health and Biomedical Research (ISABIAL, Group 23), Balmis Research Group in History of Science, Health Care and Food, Alicante, Spain, 10 Nursing and Pysiotherapy Department and Grup d'Estudis Societat, Salut, Educació i Cultura (GESEC), University of Lleida, Lleida, Spain, 11 Lleida Institute for Biomedical Research Dr. Pifarré Foundation, Health Care Research Group (GRECS), IRBLLEIDA, Lleida, Spain, 12 Diet Department, University Hospital of Navarra, Pamplona, Spain, 13 Faculty of Pharmacy and Nutrition, University of Navarra, Pamplona, Spain, 14 Surgery Unit of the Complejo Hospitalario Universitario Insular Materno Infantil de Gran Canaria, Las Palmas de Gran Canaria, Spain, 15 Nursing Department, University of Las Palmas de Gran Canaria, Las Palmas, Spain, 16 Hospital Universitario de Gran Canaria Dr. Negrín, Las Palmas, Spain, 17 Department of Nursing University of Alicante, Alicante Institute for Health and biomedical Research (ISABIAL, Group 23), Nursing Information Systems Unit Dr Balmis General University Hospital, Alicante, Spain, 18 Nursing Department, University of Murcia, Murcia, Spain, 19 Endocrinology and Diet Department, Hospital General Universitario Reina Sofía de Murcia, Murcia, Spain, 20 Research Unit, Hospital Clínico Universitario Lozano Blesa, Zaragoza, Spain, 21 Instituto de Investigación Sanitaria Aragón (IIS Aragón),. GIIS081-Care Research Group, Zaragoza, Spain, 22 Continued Training Unit, Hospital Clínico Universitario Lozano Blesa, Zaragoza, Spain, 23 Department of comunity Nursing, Research Unit for the Analysis of Mortality and Health Statistics, Preventive Medicine, Public health and History of Science, University of Alicante, Alicante, Spain, 24 Endocrinology Department, Dr. Balmis General University Hospital, Institute for Health and Biomedical Research (ISABIAL, Group 20), Alicante, Spain, 25 Department of Nursing University of Alicante, Dr Balmis General University Hospital, Alicante Institute for Health and Biomedical Research (ISABIAL, Group 20), Alicante, Spain, 26 Institute Murciano for Biosanitary Research (IMIB), Nursing and Healthcare Research Unit ENFERAVANZA, El Palmar, Murcia, Spain, 27 Research Ethics Committee UM and University General Hospital Reina Sofía, Murcia, Spain, 28 Evaluation Unit of the Canary Islands Health Service, Tenerife, Spain, 29 Red de Agencias de Evaluación de Tecnologías Sanitarias del Ministerio de Sanidad, Madrid, Spain, 30 GREISSEC Grupo Español de Investigación de Cuidados en Servicios de Salud en Enfermedades Crónicas, Unidad de Investigación en Cuidados de Salud (Investén-isciii), Madrid, Spain, 31 Area Manager of Outpatient Care (Day Hospital), Hospital Universitario la Fe, Valencia, Spain, 32 Red de Investigación en Cronicidad, Atención Primaria y Promoción de la Salud (RICAPPS), Barcelona, Spain

* isabel.orts@ua.es



**Data Availability Statement:** No datasets were generated or analyzed during the current study. All

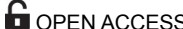

relevant data from this study will be made available upon study completion.

**Funding:** This study has been funded by Instituto de Salud Carlos III (ISCIII) through the project "PI21/00738" and co-funded by the European Union. The funders did not and will not have a role in study design, data collection and analysis, decision to publish, or preparation of the manuscript.

**Competing interests:** The authors have declared that no competing interests exist.

# Abstract

## Background

Malnutrition is a recurring problem that has become more relevant in recent years. The aim of this study is to assess the risk of malnutrition and nutritional status on admission and its evolution until discharge in patients aged 65 and over admitted to medical and surgical hospitalization units in hospitals of the Spanish National Health System.

## Methods

Prospective observational study to be carried out in the medical-surgical hospitalization units of 9 public hospitals between 01/09/2022 and 31/12/2024. Using consecutive sampling, a total of 4077 patients will be included (453 in each hospital). Variables included are related to the care process, functionality, cognition and comorbidity, risk profile, nutritional status and dysphagia; as well as frailty, dietary quality and contextual variables. The incidence of risk of malnutrition, undernutrition and dysphagia during the care process and at discharge will be calculated. The association with risk factors will be studied with logistic regression models and multivariate Cox regression models. In addition, an analysis of participants' satisfaction with food services will be carried out. The study was approved by the Ethics and Research Committee on 30/09/2020, approved for funding on 02/12/2021 and with registration number RBR-5jnbyhk in the Brazilian clinical trials database (ReBEC) for observational studies.

## Discussion

Some studies address nutritional status or dysphagia in older people in various care settings. However, there is a lack of large sample studies including both processes of the impact of hospitalization. The results of the project will provide information on the incidence and prevalence of both pathologies in the study subjects, their associated factors and their relationship with the average length of stay, mortality and early readmission. In addition, early detection of a problem such as malnutrition related to the disease and/or dysphagia during a hospital stay will favor the action of professionals to resolve both pathologies and improve the health status of patients.

## Introduction

Malnutrition is a recurring problem that has become more relevant in recent years and people aged 65 years and older are common at all levels of health care. Moreover, malnutrition can be aggravated by the presence of diseases and problems such as dysphagia [1]. In particular, assessment and monitoring of nutritional status and dysphagia during hospitalization processes are crucial to provide the highest quality care to avoid undesirable outcomes such as susceptibility to infection, morbidity, and mortality and increased premature readmissions, as well as longer length of stay [2].

Malnutrition is the result of a deficit in the intake or assimilation of nutrients that alters body composition, leading to a decrease in physical and mental capacities, which hinders

recovery during an illness process [3]. In Europe, this problem affects approximately 30 million people and costs approximately 170 billion euros per year [4]. Disease-related undernutrition, on the other hand, is caused by inadequate nutrient and energy intake about nutritional requirements in a situation where disability and disease are at the core of the problem. This type of malnutrition is due to the bodily changes and loss of functionality that occur when there is an acute or chronic illness [5]. Furthermore, different studies show that the nutritional status of patients worsens during hospitalization [1], increasing morbidity and mortality, hospital stay, and costs for the healthcare system [6,7].

People aged 65 years and older are particularly susceptible to undernutrition and disease-related undernutrition. This is due to physiological changes during ageing, the presence of associated diseases, and the specific nutritional needs of this population (decreased activity, alterations associated with swallowing disorders or frailty, among others) [8,9]. In this regard, international studies estimate that between 35–65% of older people in hospital settings are undernourished [10,11].

In Spain, the prevalence of undernutrition in people over 65 years of age is around 10%, with a risk prevalence of 23.3% [12]. In the hospital setting, the PREDYCES (Prevalence and costs of malnutrition in hospitalized patients) study established a prevalence of disease-related undernutrition of 23.7% in the general population, increasing to 37% in patients over 70 years of age [5]. Recent studies establish a prevalence of disease-related malnutrition on admission to inpatient units of 26.9% and, in addition, 18% of patients develop malnutrition during hospital stay [13]. Some authors point to an under-diagnosis of this problem, possibly related to the lack of standards in the detection and screening of malnutrition risk [13]. For these reasons, it is estimated that at least one in three patients over 65 years of age admitted to the hospital is malnourished, with an additional cost of between 1,400–1,800 for each hospital admission of a patient at risk of disease-related malnutrition [5,14].

On the other hand, oropharyngeal dysphagia is defined as the difficulty or inability to swallow solid, semi-solid, or liquid food from the oral cavity to the stomach [3]. Dysphagia is a symptom that accompanies other diseases and can affect up to 30–40% of the population over 65 years of age and up to 44% of elderly people admitted to hospital. It is estimated that, in Europe, 80% of patients are neither diagnosed nor receive any treatment for this pathology [15]. Furthermore, dysphagia is associated with a higher prevalence of malnutrition, along with other conditions such as frailty [16], neurological diseases, or neoplasms [17]. Some studies establish a joint prevalence of undernutrition and dysphagia ranging from 3–29% [18]. However, no international studies addressing both problems together in the hospital setting were retrieved.

Some factors that may explain the high rates of disease-related malnutrition and dysphagia are the lack of means and the scarcity of information on nutrition by professionals [13], as well as the quality of the food offered in hospitals [19]. The level of acceptance of hospital food has an important relationship with the user's overall satisfaction and quality of life during admission [20]. Thus, the assessment of the quality of menus during hospital admission and patient satisfaction is considered an appropriate method of nutritional assessment in patients at risk of malnutrition and dysphagia [21].

Disease-related malnutrition and dysphagia are associated with increased hospital morbidity and mortality, length of stay, readmission rates, and associated costs [22]. Nutritional screening, early detection, and monitoring of nutritional status and dysphagia should be a fundamental part of the care plan for patients, especially in patients aged 65 years and older in inpatient units [6,7]. However, research currently places insufficient emphasis on the coexistence of malnutrition and dysphagia at the hospital level. Indeed, in Spain, some studies address the nutritional status of people aged 65 years and older in various care settings, using different definitions and screening tools [1,4,16]. Despite this fact, there is a lack of national

studies on the joint prevalence of malnutrition and dysphagia conducted on a large sample of patients and including an assessment of the risk factors associated with both processes, as well as their evolution and consequences during hospitalization.

Therefore, this project will study the evolution of nutritional status and dysphagia in patients aged 65 and over during the care process in medical and surgical hospitalization units in Spanish hospitals. In addition, sociodemographic, clinical, functional status, frailty, and healthcare context-related characteristics that may be associated with the presence of malnutrition and dysphagia will be identified. Finally, patient satisfaction with food services in the participating centers will be studied.

## Materials and methods

### Research hypothesis

The nutritional status of patients aged 65 years and over admitted to the medical and surgical inpatient units of the participating hospitals evolves unfavourably from admission to discharge and may increase the length of stay, hospital mortality, and early readmission.

### Aims

The main objective of this project is to assess the risk of malnutrition and nutritional status on admission and its evolution until discharge, in patients aged 65 and over admitted to medical and surgical units in hospitals of the Spanish National Health System. In addition, the aim is to respond to the following specific objectives:

1. To identify the incidence and prevalence of risk of malnutrition and the presence of malnutrition and associated factors on patient admission and discharge.

2. To describe the evolution of nutritional status during patient hospitalization.

3. To analyze the average length of stay, hospital mortality, and early readmission as a function of nutritional status at admission and the presence of dysphagia in the study cohort.

4. To determine the incidence and prevalence of dysphagia and associated factors on admission and discharge.

5. To determine satisfaction with the quality of patient nutrition at discharge, the type of hospital food service, and the characteristics of the hospital nutrition unit.

### Design

An observational, prospective, and multicentre study will be carried out with the participation of 9 public hospitals of the Spanish National Health System, located in 7 Autonomous Communities (Andalusia, Aragon, Catalonia, Valencia, Canary Islands, Murcia, and Navarre). The study will be carried out between 01/09/2022 and 31/12/2024.

### Sample/Participants

The study population consists of the users of the medical and surgical inpatient units of the participating hospitals. Participants will be selected through consecutive case sampling.

The study will include patients aged 65 years or older who are admitted to the inpatient units, with an expected stay of at least 48h and who sign the informed consent form (or their legal guardian). Patients admitted with enteral nutrition or parenteral nutrition, terminally ill patients, patients with a primary diagnosis of cancer, patients admitted to critical care units,

and with a diagnosis of COVID-19 illness at admission or during the stay will be excluded. Withdrawal criteria during follow-up will include: i) patients transferred to another hospital or unit not participating in the study; ii) any patient meeting any exclusion criteria at any time.

To calculate the sample size, the Freeman criterion for multivariate logistic regression analysis in epidemiological studies was considered first [23]. Taking into account that a maximum of 20 independent variables (k) will be included in the model at the same time and that it is intended to carry out both a global analysis and an analysis by a hospital, an initial sample size of n = 10*(k+1) = 10*(20+1) = 210 patients per hospital is estimated and, estimating possible losses of 15%, a total of 242 patients per hospital will be necessary. Secondly, given that we intend to study the prevalence of malnutrition and/or dysphagia in each of the participating hospitals and that in the literature the prevalence of malnutrition and/or dysphagia presents great variability, it seems appropriate to consider a prevalence of 50% (will guarantee the necessary sample size) which, for the confidence of 95%, a precision of 5% and a percentage of losses of 15%, will require a sample size per hospital of 453 patients (GRANMO-IMIM sample size calculator). Therefore, the total sample will be 4077 participants, and a recruitment and follow-up period of 18 months is estimated to be necessary.

## Variables and instruments

The study variables are organized into eight groups: i) Nutritional status; ii) Dysphagia; iii) Food quality and environment; iv) Socio-demographic variables; v) Clinical, functional, cognitive, and frailty variables; vi) Variables related to the risk profile; vii) Context variables; viii) Variables related to the care process.

Specifically, nutritional status and dysphagia are considered the main variables of the study. Likewise, the variables related to nutritional status are organized into:

i.  Risk of malnutrition on admission: Malnutrition Universal Screening Tool (MUST) will be used to determine the risk of malnutrition. This questionnaire will measure current weight (kg), estimated weight three months ago (kg), height (cm), and Body Mass Index (BMI) (kg/m2). If height and weight cannot be measured or obtained, they will be estimated mid-upper arm circumference (MUAC) and calf circumference (CC) are set for weight and fore-arm length for height by the nutrition staff of each hospital. All patients included in the study will be assessed on admission, every 7 days, and at discharge.

ii.  Diagnosis of malnutrition and degree of severity at admission: Global Leadership Initiative on Malnutrition (GLIM) criteria will be used for the population identified as high risk according to MUST. In addition, the following biochemical parameters will be collected: albumin, prealbumin CRP, CRP/prealbumin ratio, transferrin, total cholesterol, and the total number of lymphocytes. They will be collected on admission, every 7 days, and discharge provided that the minimum time for their re-determination has elapsed.

iii.  Type of nutritional care established (according to risk and diagnosis of malnutrition): Includes aspects such as referral to the nutrition and dietetics unit, dietary adaptation according to alteration, nutritional supplementation, type of nutrition, or nutritional education. It will be recorded within 48 hours of admission, every 7 days and at discharge, and in case of any modification in the selected intervention, the starting date should be noted.

iv.  Daily intake: a semi-quantitative questionnaire will be used to record the amount of intake made during the days of admission.

The variables related to dysphagia include the determination of suspected dysphagia with the EAT-10 questionnaire. In the case of suspicion, a clinical examination will be carried out

to diagnose dysphagia and classification with the MECV-V questionnaire [24]. Likewise, the type of care received will be recorded according to the presence and type of dysphagia. Both instruments will be used at admission and discharge of all patients included in the study. Patients with a prolonged stay will be re-evaluated every 7 days.

In addition, the study includes a wide variety of variables related to the care process, functional status, cognitive status and comorbidities, frailty, dietary quality, and context variables. Likewise, multiple instruments will be used depending on the type of variable and other complementary instruments will be used to collect the most relevant socio-demographic and clinical data.

## Data collection

In each participating hospital, there will be a site manager and a research team consisting of nurses, nutritionists, doctors, speech therapists, endocrinologists, and technical staff. The principal investigator of the project (MIOC) will coordinate the centre managers. Each figure will be assigned specific roles related to data security and ethical issues. In particular, the site managers will be responsible for the coordination and supervision of the recruitment process and data collection in their respective hospitals. Data collection will be adapted to the specificities and material and human resources of each of the centres collaborating on the project. Fig 1 shows a flow chart of how the process of recruiting participants will be carried out, as agreed by all the participating centres.

In each centre, a researcher will undertake the daily recruitment of participants by reviewing the hospital admission register and electronic medical records to verify compliance with the selection criteria. If the patient meets the selection criteria, this investigator will locate the patient's admission unit, inform the patient personally and in detail about the study (delivery of the study information sheet) and invite the patient to participate, obtaining written informed consent. He/she will then inform the data collection coordinator and the centre manager.

The data collection process will be grouped into three phases (Fig 2).

**Phase 1: Baseline situation.** At the beginning of data collection, the facility manager will be responsible for recording the characteristics of the hospital nutrition unit and the hospital food service.

In the first 24–48 hours of admission, the nursing and nutrition staff will start collecting data related to functional variables (Barthel Index), clinical variables (diagnosis; renal or liver disease), cognitive variables (Pfeiffer questionnaire), frailty (polymedication; Frail questionnaire; Charlson index), risk profile (Braden scale; active infection), socio-demographic (Gijón scale), hospitalization and context (hospital; hospitalization unit), anthropometric parameters (height; weight; BMI), dysphagia-related variables (EAT-10; MECV-V) and malnutrition (biochemical profile; MUST; GLIM criteria).

Specifically, if the patient obtains a result $\geq 2$ in the MUST questionnaire, the GLIM criteria will be applied by the nutritionists/endocrinologists to stratify the severity of malnutrition and the type of nutritional care will be adapted. On the other hand, if the patient obtains a result $\geq 3$ in the EAT-10 questionnaire, an enter consultation with a speech therapist and endocrinologist or nutritionist will be requested for an in-depth assessment of the dysphagia with the MECV-V questionnaire and to adapt the nutritional care. In addition, the team of nutritionists at each centre will provide and explain to the patient or their companion the daily intake record booklet, which the patient or guardian must fill in during their hospital stay and which will be supervised in the morning shift by the nurse or nutritionists participating in the study. Similarly, within the first 24 hours of admission, the person responsible for the study in

## Figure 1.

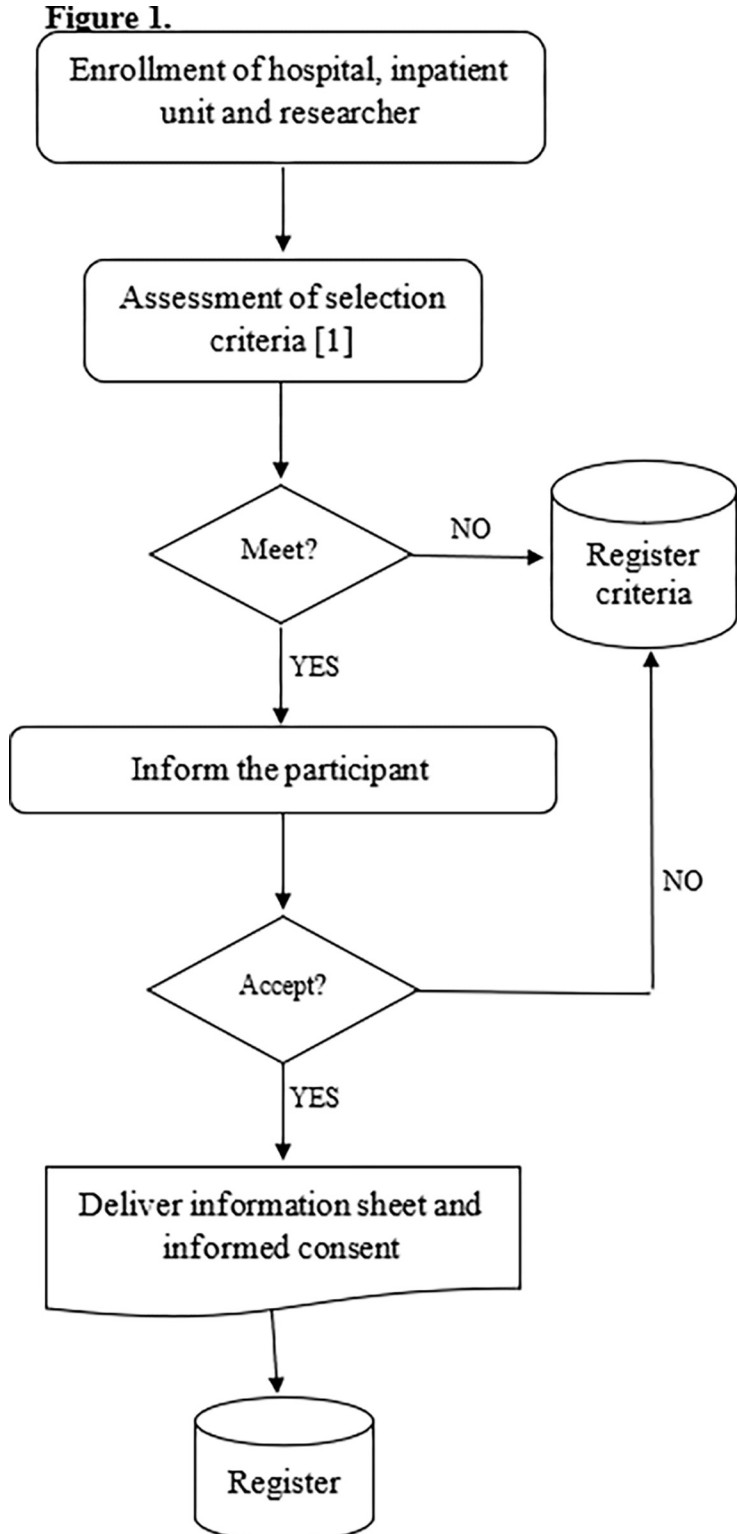

[1] SELECTION CRITERIA

**Inclusion:**
Patients aged 65 years and older admitted to inpatient units selected in each hospital
Hospital stay greater than 48 hours.
Informed consent signature

**Exclusion, patients:**
with enteral or parenteral nutrition on admission
in a terminal situation or with a main diagnosis of cancer
admitted to intensive care units
entering or developing the disease with COVID-19 diagnosis

**Fig 1. Flowchart of the recruitment of participants.**

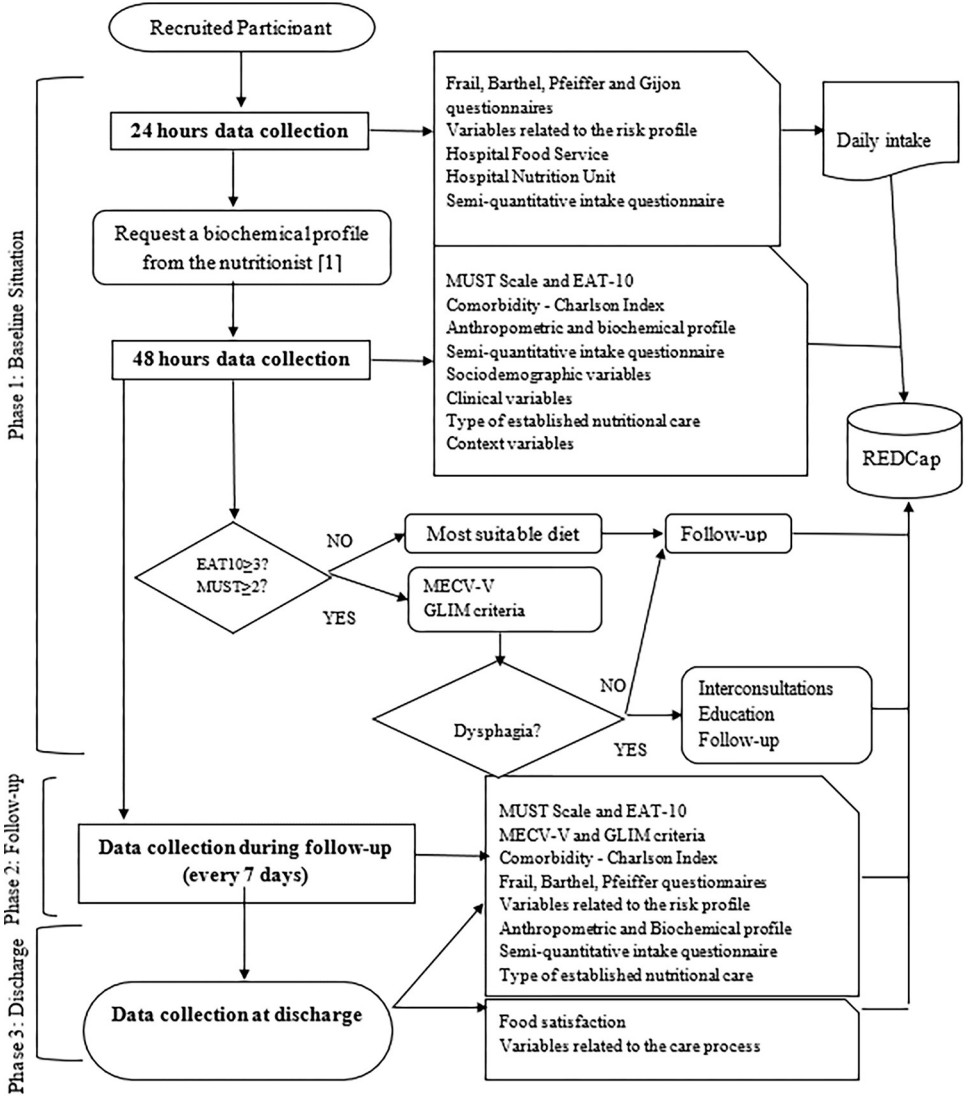

**Fig 2. Flowchart of the data collection.**

each hospital will contact the medical team to add the biochemical profiles (albumin, pre-albumin CRP, CRP/prealbumin ratio, transferrin, total cholesterol, and the total number of lymphocytes).

**Phase 2: Follow-up.** Participants will be monitored every 7 days until they are discharged from the hospital. In this way, the researchers in charge will make a forecast of follow-up dates. In addition to the variables indicated, the nutritional care established will be adapted according to the evolution of the nutritional status and degree of dysphagia of the patient during their hospital stay, as part of the usual practice in each centre. On each occasion, a record will be kept of the person collecting the data.

**Phase 3: Situation on discharge.** In addition to the variables indicated, on the day of discharge from the hospital, the daily intake record book will be collected and the food satisfaction survey will be provided. The variables related to the care process (days of stay, hospital mortality, and whether it was early readmission) will also be recorded.

Finally, the patient will be given the continuity of care report at discharge, drawn up by the responsible physician together with the nursing recommendations and information sheets on care related to malnutrition or dysphagia, as appropriate.

Data collection will be carried out electronically through Research Electronic Data Capture (REDCap) (www.prooject-redcap.org) [25,26]. This is a web application that allows the creation of electronic data notebooks for research projects and their online and offline collection from computers and other mobile devices. It also includes a scheduling and alarm system to manage data collection during monitoring. In this regard, it should be noted that the researchers who take on the data collection will receive training on the operation of REDCap and that a pilot of the process of recruiting participants and data collection will be carried out for one month.

## Data analysis

A descriptive analysis of all the study variables will be carried out, using the appropriate measures according to the type of variable and their corresponding 95% CIs. Firstly, depending on the groups to be compared and compliance with the application requirements, a bivariate analysis will be performed, including the chi-squared or Fisher's exact test (qualitative variables) and Student's t-test, ANOVA, Mann-Whitney U-test, or Kruskal-Wallis (quantitative variables), depending on the application conditions.

For the analysis of the evolution of the risk of malnutrition, McNemar's test or Cochran's Q test will be used for qualitative variables and paired t-test or Friedman's rank test for quantitative variables. The incidence of risk of malnutrition and undernutrition will be calculated, as for dysphagia, during the care process and at patient discharge, estimating their corresponding Odds Ratios and 95% CI using logistic regression models.

To assess the possible risk factors for both malnutrition and dysphagia and the socio-demographic and clinical characteristics of the patients, multivariate logistic regression models will be constructed. As independent variables will include: type of nutritional care established according to the risk of malnutrition; daily intake; suspected and diagnosed dysphagia and days of stay. As control variables will include: those related to the risk profile; sociodemographic (sex, age); clinical, functional, cognitive and frailty variables. In the case of mortality and time to readmission, multivariate Cox regression models will be fitted. Independent variables will be included: level of risk of malnutrition at admission; diagnosis of malnutrition at admission; change in risk of malnutrition at admission, at 7 days and at discharge; change in nutritional status at admission, at 7 days and at discharge. Control variables will include: type of nutritional care established (according to risk of malnutrition); daily intake; contextual; related to risk profile; sociodemographic; clinical, functional, cognitive and frailty variables; suspected and diagnosed dysphagia. In all multivariate models, possible interaction effects and possible collinearity will be analyzed.

Additionally, multilevel models (generalized linear mixed models) with multivariate logistic link function will be implemented to account for possible correlation between patients admitted to the same medical or surgical units (first level) or hospital (second level).

In case of losses to follow-up, multiple imputation techniques will be used for missing data. In all cases, bilateral contrast will be used and will be considered significant when $p < 0.05$. The analysis will be performed using SPSS v28.

## Limitations, validity, and reliability

The multicentre nature of the study may affect the validity and reliability of the information collected in the development of the study, although it is necessary to collaborate with different

centres to achieve a large sample size and a geographical distribution of centres that can provide an overview of the situation of this important problem in our context.

The strategies developed to control possible selection and information biases derived from the multicentre nature of the study were described in the text and are related to the development of a procedure for patient recruitment and data collection that has been agreed upon by all the participating centres, taking into account their idiosyncrasies. In addition, this procedure will be piloted for one month, before the start of data collection.

Another strategy consists of the use of REDCap, as it will allow data collection to be carried out directly on electronic devices and transferred to a single database, avoiding possible errors derived from the transcription of data and the unification of different databases. However, it will be necessary to develop an agreed data notebook, train researchers in the use of the platform, and, as mentioned above, carry out a pilot test. In addition, REDCap includes an alert system that will facilitate the organization of patient follow-up every 7 days, reducing the risk of data loss.

Another aspect that may affect the validity and reliability of the information is related to a large number of variables and measurement instruments used in the study and the variability of the data collected in the electronic medical records of each centre as they use different programs. To minimize the possible consequences, those responsible for each centre and the principal investigator agreed in previous meetings on the variables to be included in the study and selected the measurement instruments based on their psychometric properties, diagnostic capacity, and international recommendations on screening for malnutrition and dysphagia.

## Discussion

Malnutrition, in people aged 65 years and older, is a common problem at all levels of health care, from primary to specialized care and in geriatric care centres. Dysphagia is one of the main disorders associated with malnutrition. Monitoring nutritional status and dysphagia during hospitalization is crucial to provide the highest quality care in the context of specialized care, with repercussions on the continuity of care in the primary care setting, at home, or in social and health care institutions.

Assessing the impact of hospitalization and establishing an individualized nutritional care plan will make it possible to recover or maintain the nutritional status appropriate to the needs of each patient. This will require the coordination of a group of professionals from different healthcare areas with a care and management profile to perform a quality action adapted to the needs of each person during hospitalization. The assessment by a multidisciplinary team is necessary for the diagnosis, supervision, and treatment of patients with malnutrition and suspected or diagnosed dysphagia. The significance of the multidisciplinary team does not remain in individual specificity but more important factors. The main objective of these teams is to offer the highest quality given their characteristics and a joint work focused on the patient, quality standards, the best diagnostic expectations, and therapeutic possibilities supported by scientific evidence, thus achieving a correct use of healthcare incomes, greater efficiency, lower risk for the patient, improved continuity of care and greater user satisfaction [27]. For this reason, the development of this project will show the relevance of multidisciplinary teams in this care and the use of organized and systematized work circuits.

Unlike previous studies [5,11,14] the NUTRIFAG Project introduces the concept of satisfaction during the hospital process. By assessing the quality of food and adapting dietary needs during admission. In addition, a common circuit adapted to the different participating hospitals in Spain is established in terms of individualized nutritional care according to the associated pathology, malnutrition, dysphagia, or both. It shows an example of updated and relevant working circuits for improvements in the care provided to patients during hospitalization.

This project will help to understand the different contexts of action, at the hospital level, and identify the variables that influence the development of malnutrition and dysphagia.

To propose the creation of future research that evaluated the effectiveness of screening and management of malnutrition and dysphagia and lead to the development of detection and intervention protocols for the management of both problems in the hospital setting. In this way, a new approach will be implemented in the planning of quality care for the patient/family, thus reducing the substitution of performance in clinical practice.

## Conclusions

This article presents the research protocol of the NUTRIFAG Project to make dysphagia and malnutrition visible as current problems in Spanish hospitals. The results of the project will provide relevant information on the incidence and prevalence of both pathologies, their associated factors, and their relationship with the average length of stay, hospital mortality, and early readmission.

In addition, the results of this project will provide information on the evolution of nutritional status and dysphagia during hospitalization, based on screening and follow-up during the hospital stay of people aged 65 or over, which can be included in protocols and clinical practice guidelines. Facilitating the inclusion of the assessment of food satisfaction during hospital admission as a novelty in the evaluation of patient quality during an acute process. These results will serve as the basis for the development of future research that evaluates nutritional crises related to malnutrition and dysphagia, adapted to the clinical situation of the patients.

## Acknowledgments

We are indebted to all the professionals who have collaborated in the project, especially the nursing staff and nutritionists of the hospitals where the study will be carried out.

NUTRIFAG team (in alphabetical order): Ángel Luís Abad-González, Marco Aldonza-Torres, Purificación Alemany-Soler, Elena Altarribas-Bolsa, María del Carmen Amoedo-Albero, María Argente-Pla, Julia Arregui-Eslava, Esther Barrufet-Alcántara, Joan Blanco-Blanco, Gema Buendía-Jiménez, Rafaela Camacho-Bejarano, Francisco Javier Carrasco-Sánchez, Cristina Carretero-Randez, Jenifer Castellano-Santana, Pedro Raúl Castellano-Santana, Mª Trinidad Castillo-García, Marta Charlo-Bernardos, Esperanza Ciérvide-Górriz, Rosa Ana Clement-Santamaría, Inmaculada Dapena-Romero, Clara De la Fuente-Gómez, Ana María De Pascual y Medina, Manuela Domingo-Pozo, Cristina Domínguez-Gadea, Nerea Elizondo-Rodríguez, Marta Escribano-García, Susana Fernández-Carrasco, Marina Figueras-Acebillo, Ascensión Franco-Bernal, José Ángel Franco-Romero, Gloria Fréyer-Rodríguez, Alicia Gainza-Calleja, Eva María Gascó-Santana, Antonia Gomariz-Martínez, María Bienvenida Gómez-Sánchez, Víctor Manuel González-Chordá, Himar González-Pacheco, Josep María Gutiérrez-Vilaplana, Beatriz Herrero-Cortina, Cristina Hurtado-Soler, Guadalupe de la Peña Jaldón-Hidalgo, Érika María Lorenzo-Ramos, Amando Márquez-Sixto, Patricia Martín-Romaní, Silvia Martín-Sanchís, Idaira Martín-Santana, Jordi Martínez-Soldevila, María Mateo-Polo, Laura Meseguer-Galiana, Andrea Micó-García, Marzena Mikla, Joaquín Moncho, Mercedes del Pilar Montalván-Pelegrín, Jesica Montero-Marco, Mª José Morano-Torrescusa, Juan María Morillas-Ruiz, Mª Julia Ocón-Bretón, María Isabel Orts-Cortés, Paloma Portillo-Ortega, Vanesa Ramos-Abril, Isabel Rebollo-Pérez, Mariona Rocaspana-García, Felipe Rodríguez-De Castro, Beatriz, Rodríguez-Ojeda, Margarita Rodríguez-Pérez, Aránzazu Ruiz-Heras-Hera, Lourdes Salinero-González, Ana Belén Sánchez-García, Mª Dolores Santos-Rey, Carolina Sorolla-Villas, Mónica Timoneda-Company, Eva María Trescastro-López, Antonia Inmaculada Zomeño-Ros, Ginesa Zomeño-Ros.

## Author Contributions

**Conceptualization:** Cristina Carretero-Randez, María Isabel Orts-Cortés, Margarita Rodríguez-Pérez, Víctor Manuel Gonzalez-Chordá, Eva María Trescastro-López, Rafaela Camacho-Bejarano.

**Formal analysis:** Cristina Carretero-Randez, María Isabel Orts-Cortés, Margarita Rodríguez-Pérez, Víctor Manuel Gonzalez-Chordá, Eva María Trescastro-López, Rafaela Camacho-Bejarano.

**Funding acquisition:** Cristina Carretero-Randez, María Isabel Orts-Cortés, Margarita Rodríguez-Pérez, Víctor Manuel Gonzalez-Chordá, Eva María Trescastro-López, Rafaela Camacho-Bejarano.

**Investigation:** Cristina Carretero-Randez, María Isabel Orts-Cortés, Margarita Rodríguez-Pérez, Víctor Manuel Gonzalez-Chordá, Eva María Trescastro-López, Rafaela Camacho-Bejarano.

**Methodology:** Cristina Carretero-Randez, María Isabel Orts-Cortés, Margarita Rodríguez-Pérez, Víctor Manuel Gonzalez-Chordá, Eva María Trescastro-López, Joan Blanco-Blanco, Jordi Martínez-Soldevila, Aránzazu Ruiz-Heras-Hera, Pedro Raúl Castellano-Santana, Amando Marquez-Sixto, Manuela Domingo-Pozo, Antonia Inmaculada Zomeño-Ros, Jesica Montero-Marco, Marta Charlo-Bernardos, Joaquín Moncho, Ángel Luís Abad-González, María Trinidad Castillo-García, Ana Belén Sánchez-García, Ana María De Pascual y Medina, Rosa Ana Clement-Santamaría, Ascensión Franco-Bernal, Rafaela Camacho-Bejarano.

**Project administration:** María Isabel Orts-Cortés.

**Resources:** Cristina Carretero-Randez, María Isabel Orts-Cortés, Margarita Rodríguez-Pérez, Víctor Manuel Gonzalez-Chordá, Eva María Trescastro-López, Joan Blanco-Blanco, Jordi Martínez-Soldevila, Aránzazu Ruiz-Heras-Hera, Pedro Raúl Castellano-Santana, Amando Marquez-Sixto, Manuela Domingo-Pozo, Antonia Inmaculada Zomeño-Ros, Jesica Montero-Marco, Marta Charlo-Bernardos, Joaquín Moncho, Ángel Luís Abad-González, María Trinidad Castillo-García, Ana Belén Sánchez-García, Ana María De Pascual y Medina, Rosa Ana Clement-Santamaría, Ascensión Franco-Bernal, Rafaela Camacho-Bejarano.

**Software:** Cristina Carretero-Randez, María Isabel Orts-Cortés, Margarita Rodríguez-Pérez, Víctor Manuel Gonzalez-Chordá, Eva María Trescastro-López, Joan Blanco-Blanco, Jordi Martínez-Soldevila, Aránzazu Ruiz-Heras-Hera, Pedro Raúl Castellano-Santana, Amando Marquez-Sixto, Manuela Domingo-Pozo, Antonia Inmaculada Zomeño-Ros, Jesica Montero-Marco, Marta Charlo-Bernardos, Joaquín Moncho, Ángel Luís Abad-González, María Trinidad Castillo-García, Ana Belén Sánchez-García, Ana María De Pascual y Medina, Rosa Ana Clement-Santamaría, Ascensión Franco-Bernal, Rafaela Camacho-Bejarano.

**Supervision:** Cristina Carretero-Randez, María Isabel Orts-Cortés, Margarita Rodríguez-Pérez, Víctor Manuel Gonzalez-Chordá, Eva María Trescastro-López, Joan Blanco-Blanco, Jordi Martínez-Soldevila, Aránzazu Ruiz-Heras-Hera, Pedro Raúl Castellano-Santana, Amando Marquez-Sixto, Manuela Domingo-Pozo, Antonia Inmaculada Zomeño-Ros, Jesica Montero-Marco, Marta Charlo-Bernardos, Joaquín Moncho, Ángel Luís Abad-González, María Trinidad Castillo-García, Ana Belén Sánchez-García, Ana María De Pascual y Medina, Rosa Ana Clement-Santamaría, Ascensión Franco-Bernal, Rafaela Camacho-Bejarano.

**Writing – original draft:** Cristina Carretero-Randez, María Isabel Orts-Cortés, Margarita Rodríguez-Pérez, Víctor Manuel Gonzalez-Chordá, Eva María Trescastro-López, Joan Blanco-Blanco, Jordi Martínez-Soldevila, Aránzazu Ruiz-Heras-Hera, Pedro Raúl Castellano-Santana, Amando Marquez-Sixto, Manuela Domingo-Pozo, Antonia Inmaculada Zomeño-Ros, Jesica Montero-Marco, Marta Charlo-Bernardos, Joaquín Moncho, Ángel Luís Abad-González, María Trinidad Castillo-García, Ana Belén Sánchez-García, Ana María De Pascual y Medina, Rosa Ana Clement-Santamaría, Ascensión Franco-Bernal, Rafaela Camacho-Bejarano.

**Writing – review & editing:** Cristina Carretero-Randez, María Isabel Orts-Cortés, Margarita Rodríguez-Pérez, Víctor Manuel Gonzalez-Chordá, Eva María Trescastro-López, Joan Blanco-Blanco, Jordi Martínez-Soldevila, Aránzazu Ruiz-Heras-Hera, Pedro Raúl Castellano-Santana, Amando Marquez-Sixto, Manuela Domingo-Pozo, Antonia Inmaculada Zomeño-Ros, Jesica Montero-Marco, Marta Charlo-Bernardos, Joaquín Moncho, Ángel Luís Abad-González, María Trinidad Castillo-García, Ana Belén Sánchez-García, Ana María De Pascual y Medina, Rosa Ana Clement-Santamaría, Ascensión Franco-Bernal, Rafaela Camacho-Bejarano.

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
