## [Decision Letter · Decision Letter 0]

24 May 2023

PONE-D-23-04586Impact of hospitalization on nutritional status and dysphagia in persons aged 65 years and over (NUTRIFRAG Study): protocol for a prospective observational studyPLOS ONE

Dear Dr. Orts Cortés,

Thank you for submitting your manuscript to PLOS ONE. After careful consideration, we feel that it has merit but does not fully meet PLOS ONE’s publication criteria as it currently stands. Therefore, we invite you to submit a revised version of the manuscript that addresses the points raised during the review process.

 Thank you for your submission. Please address reviewers' comments and will be happy to consider a revised version for publication.

We look forward to receiving your revised manuscript.

Kind regards,

Mario Ulises Pérez-Zepeda, M.D., Ph.D.

Academic Editor

PLOS ONE

Journal Requirements:

Reviewers' comments:

Reviewer's Responses to Questions

**Comments to the Author**

1. Does the manuscript provide a valid rationale for the proposed study, with clearly identified and justified research questions?

Reviewer #1: Yes

Reviewer #2: Partly

2. Is the protocol technically sound and planned in a manner that will lead to a meaningful outcome and allow testing the stated hypotheses?

Reviewer #1: Yes

Reviewer #2: Partly

3. Is the methodology feasible and described in sufficient detail to allow the work to be replicable?

Reviewer #1: Yes

Reviewer #2: Yes

4. Have the authors described where all data underlying the findings will be made available when the study is complete?

Reviewer #1: Yes

Reviewer #2: Yes

5. Is the manuscript presented in an intelligible fashion and written in standard English?

Reviewer #1: Yes

Reviewer #2: Yes

6. Review Comments to the Author

You may also provide optional suggestions and comments to authors that they might find helpful in planning their study.

Reviewer #1: Abstact

Malnutrition is a recurrent problem that has acquired its own identity in recent years : make this statement more clear, what do u mean with own identity.

Objective:

‘’assess the risk of malnutrition and nutritional status on admission and its evolution until discharge’’ and the dysphagia? Doesnt go with the title

Why is it registered in brazi? registration number RBR-5jnbyhk in the Brazilian clinical trials database (ReBEC) for observational studies.

‘’These results will serve to improve their health status’’: cold you be less general?

180. Still confused with the title what is the role of the dysphagia there; 181 Research hypothesis

the main aim is not something new, i would have expected they testing and intervention to prevent inhospital malnutrition or to find risk markers for dysphagia and malnutrition

421. 431 ''This project will help to understand the different contexts of action (at specialized care and hospitalization unit level) to propose the creation of an evidence-based protocol that allows for comprehensive and multidisciplinary care of malnutrition and dysphagia. In this way, a new approach to planning quality care for the patient/family will be implemented, thereby reducing the variability of action in clinical practice''

How is this project going to extrapolate to the creation of protocols. Protocol of detection? Are they in the objetives testing an instruments? Ot protocolo for intervention? Are they testing an intervention? The study is very Good, well design and they are collecting valuable information. hOWEVER what they Will do with that information is blurry, jsut to see who develops malnutrition’’ and what with that ?

437. ''In the conclusions: evidence on the usefulness of screening and monitoring malnutrition and dysphagia during hospital admission of people aged 65 and over, ''

That conclusion does not reflected in any of the objectives

446.'' These results will serve as a basis for the development and evaluation of nutritional interventions adapted to the presence of malnutrition and dysphagia and above all to the clinical situation of the patients. ''

How?, using the dietary records that you are measuring? Measuring textures for those with dysphagia ???

Following the objetives again this project cant do any of this, the main objective is to asses malnutrition incidence, but from that to these conclusión there is a huge gap

Reviewer #2: This study protocol entitled “Impact of hospitalization on nutritional status and dysphagia in persons aged 65 years and over (NUTRIFRAG Study): protocol for a prospective observational study” explored the issue of malnutrition, dysphagia, and associated outcomes in the elderly inpatients in Spain. Even this topic is worth to explore, several points needed to be stated more clearly in advance.

First, the title stated that NUTRIFRAG Study is a prospective observational study. For observational mostly means that the physician and medical team will do nothing when disclosing malnutrition or dysphagia, no nutrient suggestion, no speech therapy, neither medication adjustment. This study is indeed a prospective study, and the term of “observational” might not be suitable for description. It was suggested to modify this term for that all inpatient would receive standardized medical care during hospitalization ethically. While food preparation and management looks to be an intervention, therefore not so "observational" in methodology.

Second, for body weight change playing a critical role on calculating the nutitrional score in previous 7 days, there is no adequate explanation on avoiding possible bias among medical inpatient with fluid overload (weight loss due to treatment in hospitalization), or in surgical inpatient receiving organ resection (weight loss) or prosthesis implantation (weight gain).

Third, it was stated that patient meeting any exclusion criteria at any time would be withdrawal from analysis. If patient with severe dysphagia received enteral nutrition by feeding tube, will the patient be removal from analysis? For tube feeding being a clinical management for dysphagia in general, will that patient withdrawal lead to another potentially analytic bias?

Final, we will need a clear statistical model for data analysis. For example, how to definite the reference group for comparison with your target population? What confounders will be adjusted in multivariate analysis and Cox regression model? It was encouraged to be stated clearly prior to participants’ enrollment.

7. PLOS authors have the option to publish the peer review history of their article (what does this mean?). If published, this will include your full peer review and any attached files.

Reviewer #1: No

Reviewer #2: **Yes: **Liang-Yu Chen

---

## [Author Response · Author response to Decision Letter 0]

13 Jun 2023

Reviewer 1 

Abstract. Malnutrition is a recurrent problem that has acquired its own identity in recent years : make this statement more clear, what do u mean with own identity.

Response: This statement is intended to show that malnutrition is a common problem that has become more prominent in recent years. The wording of this paragraph is modified in lines 77 and 109 to make the text easier to understand. Thank you for your comment.

Abstract. Objective: ‘’assess the risk of malnutrition and nutritional status on admission and its evolution until discharge’’ and the dysphagia? Doesnt go with the title. 

Response: Thank you for your comment. As there is no evidence that hospitalization causes or is a precursor to dysphagia, we have decided to remove the word dysphagia (line 1) from the title to maintain the consistency of this research protocol.

Abstract. Why is it registered in brazi? registration number RBR-5jnbyhk in the Brazilian clinical trials database (ReBEC) for observational studies.

Response: We have registered in the Brazilian clinical trials database because we were informed in detail about the publication of research protocols. Therefore, we registered our project on this platform, as it is an international database, which will favor the dissemination of our project.

Abstract. ‘’These results will serve to improve their health status’’: cold you be less general?

Thank you very much for your clarification. This statement refers to the fact that detecting a problem such as malnutrition related to illness and dysphagia during a hospital stay will help professionals to solve these pathologies. Considering that feeding and the correct swallowing process are physiological needs of the human being that must be preserved daily to favor an optimal quality of life, this statement has been modified in lines 99-102 to improve its understanding.

180. Still confused with the title what is the role of the dysphagia there; 181 Research hypothesis

Response: Thank you for your comment. As there is no evidence that hospitalization causes or is a precursor to the development of dysphagia, we have decided to remove the word dysphagia from the title to maintain the consistency of this research protocol.

The main aim is not something new, i would have expected they testing and intervention to prevent in hospital malnutrition or to find risk markers for dysphagia and malnutrition

Response: As described in the introduction, few studies in Spain provide data on malnutrition and dysphagia in a significant number of hospitals. Therefore, before considering an experimental study or an intervention, we needed to know the current status of the subject at the national level, adapted to the context of each hospital. For this reason, we first designed an observational study to, in the future, develop studies that would allow us to analyze interventions to prevent and/or treat both pathologies. We share the same idea, but initial data are needed to assess variability in different hospital contexts.

421. 431 ''This project will help to understand the different contexts of action (at specialized care and hospitalization unit level) to propose the creation of an evidence-based protocol that allows for comprehensive and multidisciplinary care of malnutrition and dysphagia. In this way, a new approach to planning quality care for the patient/family will be implemented, thereby reducing the variability of action in clinical practice''

How is this project going to extrapolate to the creation of protocols. Protocol of detection? Are they in the objectives testing an instruments? Or protocol for intervention? Are they testing an intervention? The study is very Good, well design and they are collecting valuable information. However what they Will do with that information is blurry, jsut to see who develops malnutrition’’ and what with that?

Response: Thank you very much for your comment and for highlighting the value of the study. We understand that this text lacks the information refuted in the previous response. 

From the comparison of the variability in care, by the hospital, about the degree of malnutrition and dysphagia, it will be possible to identify the variables that influence the development of malnutrition and dysphagia. With this information, future research will be conducted to evaluate interventions on the effectiveness of screening and management of malnutrition and dysphagia leading to the development of screening and intervention protocols for the management of both problems in the hospital setting. The text is modified to improve understanding of the protocol on lines 445-451.

437. ''In the conclusions: evidence on the usefulness of screening and monitoring malnutrition and dysphagia during hospital admission of people aged 65 and over'' That conclusion does not reflected in any of the objectives

Response: Thank you for your comment. This question has been amended on lines 459-462 of the text.

446.'' These results will serve as a basis for the development and evaluation of nutritional interventions adapted to the presence of malnutrition and dysphagia and above all to the clinical situation of the patients. ''

How?, using the dietary records that you are measuring? Measuring textures for those with dysphagia ???

Response: Thank you very much for your comment. We understand that this text is missing the information refuted in the previous answers. The text is amended on lines 464-466. 

Earlier reference was made to the comparison in the variability of care in cases of malnutrition and dysphagia. This means that when a case of malnutrition and dysphagia is detected, based on the assessment instruments indicated, the type of care received is recorded, in a conventional manner and depending on each hospital. In addition to the dietary records and the texture required by each patient, postural and hygiene education, and consultation with specialists depending on the patient's situation, among others, are contemplated. All of these are procedures established in each hospital depending on the degree of malnutrition and dysphagia of each patient. Describing the variability in the multidisciplinary actions of the working teams that intervene in the nutrition units will favor the development of future research to evaluate interventions on the effectiveness of screening and management of malnutrition and dysphagia that will lead to the development of detection and intervention protocols for the management of both problems in the hospital environment.

Following the objectives again this project can’t do any of this, the main objective is to asses malnutrition incidence, but from that to these conclusion there is a huge gap.

Response: We hope that from the responses described in your review, your understanding of the protocol has improved and will allow you to reconsider this final assessment of the protocol. Thank you very much for your comments, as they have allowed us to correctly define the purpose of the protocol.

Reviewer 2

This study protocol entitled “Impact of hospitalization on nutritional status and dysphagia in persons aged 65 years and over (NUTRIFRAG Study): protocol for a prospective observational study” explored the issue of malnutrition, dysphagia, and associated outcomes in the elderly inpatients in Spain. Even this topic is worth to explore, several points needed to be stated more clearly in advance.

First, the title stated that NUTRIFRAG Study is a prospective observational study. For observational mostly means that the physician and medical team will do nothing when disclosing malnutrition or dysphagia, no nutrient suggestion, no speech therapy, neither medication adjustment. This study is indeed a prospective study, and the term of “observational” might not be suitable for description. It was suggested to modify this term for that all inpatient would receive standardized medical care during hospitalization ethically. While food preparation and management looks to be an intervention, therefore not so "observational" in methodology.

Response: Thank you for your comment. We understand that this description is confusing and was a matter of great deliberation by the methodological group. We consider it an observational study because the research team does not decide on the intervention to be performed once the degree of malnutrition and dysphagia has been detected, nor does it control which participants receive it or not. Only the standardized procedure of action and nutritional and dysphagia care is recorded to know the variability in each of the hospitals. Taking into account the question, the study detects a nutritional or dysphagia problem and refers to specific units for care. The decision does not rest with the individual researcher. In this way, the performance of each hospital can be analyzed. We do not consider it to be an experimental study because the researcher does not manipulate the explanatory variable.

Second, for body weight change playing a critical role on calculating the nutitrional score in previous 7 days, there is no adequate explanation on avoiding possible bias among medical inpatient with fluid overload (weight loss due to treatment in hospitalization), or in surgical inpatient receiving organ resection (weight loss) or prosthesis implantation (weight gain).

Response: An explanatory document detailing the variables included were developed during the drafting of the protocol. This document could not be included in this platform due to its length. During the stay, the patient is assessed objectively, including the diagnosis based on ICD-10 coding (both on admission and discharge) and clinical variables such as renal function and liver function (during admission, follow-up, and discharge) to assess their influence on weight and associated clinical changes during the hospital stay. As well as biochemical parameters that reflect the circumstances described, such as albumin or prealbumine. 

In those surgical units that present the aforementioned situations, organ resection or prosthesis implantation, by using both conventional weight and the measurement of circumferences and lengths, the recording of intake, and analytical values, a bias concerning the rest of the patients is not contemplated.

Third, it was stated that patient meeting any exclusion criteria at any time would be withdrawal from analysis. If patient with severe dysphagia received enteral nutrition by feeding tube, will the patient be removal from analysis? For tube feeding being a clinical management for dysphagia in general, will that patient withdrawal lead to another potentially analytic bias?

Response: Thank you for your input and for allowing us to clarify this point. If a patient at the time of recruitment is on enteral nutrition (EN) and/or parenteral nutrition (PN), he/she is not included. In the case of including a patient who is not initially on EN and/or PN and during admission starts any type of nutrition, he/she will not be withdrawn from the study. The caloric intake according to the type of nutrition received shall be recorded in the intake notebook. In this way, the nutritional situation of the patient is known, avoiding possible analytical bias.

Final, we will need a clear statistical model for data analysis. For example, how to definite the reference group for comparison with your target population? What confounders will be adjusted in multivariate analysis and Cox regression model? It was encouraged to be stated clearly prior to participants’ enrollment.

Response: Thank you for your comment. This question has been amended in lines 365-379 of the text.

---

## [Editor Report · Decision Letter 1]

26 Jun 2023

Impact of hospitalization on nutritional status in persons aged 65 years and over (NUTRIFAG Study): protocol for a prospective observational study

PONE-D-23-04586R1

Dear Dr. Orts Cortés,

We’re pleased to inform you that your manuscript has been judged scientifically suitable for publication and will be formally accepted for publication once it meets all outstanding technical requirements.

Kind regards,

Mario Ulises Pérez-Zepeda, M.D., Ph.D.

Academic Editor

PLOS ONE

Additional Editor Comments (optional):

Thank you for your revised work, I think you have addressed the reviewers' concerns, so I am happy to move forward with the acceptance of your work.
---

## [Editor Report · Acceptance letter]

28 Jun 2023

PONE-D-23-04586R1 

Impact of hospitalization on nutritional status in persons aged 65 years and over (NUTRIFRAG Study): protocol for a prospective observational study 

Dear Dr. Orts Cortés:

I'm pleased to inform you that your manuscript has been deemed suitable for publication in PLOS ONE. Congratulations! Your manuscript is now with our production department. 

Kind regards, 

on behalf of

Dr. Mario Ulises Pérez-Zepeda 

Academic Editor

PLOS ONE